# (Endo)Cannabinoids and Gynaecological Cancers

**DOI:** 10.3390/cancers13010037

**Published:** 2020-12-25

**Authors:** Anthony H. Taylor, Daniel Tortolani, Thangesweran Ayakannu, Justin C. Konje, Mauro Maccarrone

**Affiliations:** 1Endocannabinoid Research Group, Reproductive Sciences Section, Department of Cancer Studies and Molecular Medicine, University of Leicester, Leicester LE1 7RH, UK; aht13@le.ac.uk (A.H.T.); thangesweran.ayakannu@lwh.nhs.uk (T.A.); 2Department of Molecular and Cell Biology, University of Leicester, Leicester LE1 7RH, UK; 3European Centre for Brain Research, IRCCS Santa Lucia Foundation, 00164 Rome, Italy; d.tortolani@alice.it; 4Gynaecology Oncology Cancer Centre, Liverpool Women’s NHS Foundation Trust, Liverpool Women’s Hospital, Liverpool L8 7SS, UK; 5Faculty of Health and Life Sciences, University of Liverpool, Liverpool L69 3GB, UK; 6Department of Biotechnological and Applied Clinical Sciences, University of L’Aquila, 67100 L’Aquila, Italy

**Keywords:** cannabinoids, cervical cancer, endocannabinoids, endometrial cancer, enzymes, receptors, gynaecological cancer, ovarian cancer, signal transduction, transport

## Abstract

**Simple Summary:**

Cancers of the female reproductive system are common and are responsible for a large number of deaths in women. The exact reasons why some of these cancers occur are unknown. It is, however, known that for most of these cancers, several factors interact for them to happen. These interactions involve factors external and internal to the woman. An understanding of some of the internal factors involved in how these cancers arise will not only help drive preventive strategies, but will speed the development of new treatment approaches. The endocannabinoid system is a family including chemicals (known as endocannabinoids) produced in the body that are similar to those derived from the *cannabis* plant. This system, which is widely distributed in the body, has been shown to be involved in various functions. Its disruption has been shown to lead to various diseases, one of which is cancer. In this review, we summarise current knowledge of this system, its various constituents, and how they are involved in reproductive events and their pathologies, especially cancers. Furthermore, we discuss the role of the endocannabinoid system in these cancers and how targeting it could lead to new approaches to diagnosis and treatment of cancers of the female reproductive system.

**Abstract:**

Gynaecological cancers can be primary neoplasms, originating either from the reproductive tract or the products of conception, or secondary neoplasms, representative of metastatic disease. For some of these cancers, the exact causes are unknown; however, it is recognised that the precise aetiopathogeneses for most are multifactorial and include exogenous (such as diet) and endogenous factors (such as genetic predisposition), which mutually interact in a complex manner. One factor that has been recognised to be involved in the pathogenesis and progression of gynaecological cancers is the endocannabinoid system (ECS). The ECS consists of endocannabinoids (bioactive lipids), their receptors, and metabolic enzymes responsible for their synthesis and degradation. In this review, the impact of plant-derived (*Cannabis* species) cannabinoids and endocannabinoids on gynaecological cancers will be discussed within the context of the complexity of the proteins that bind, transport, and metabolise these compounds in reproductive and other tissues. In particular, the potential of endocannabinoids, their receptors, and metabolic enzymes as biomarkers of specific cancers, such as those of the endometrium, will be addressed. Additionally, the therapeutic potential of targeting selected elements of the ECS as new action points for the development of innovative drugs will be presented.

## 1. Introduction

### 1.1. Cannabis and Endocannabinoids and the Discovery of the Endocannabinoid System

*Cannabis* is the botanical name of an annual herbaceous plant of the *Cannabaceae* family that is cultivated and distributed all over the world. This genus consists of three major species, *C. sativa*, *C. indica,* and *C. ruderalis*, which, through interbreeding, share similar genetic backgrounds and physical traits [1]. One distinctive trait of *Cannabis* plants is production of secondary compounds called “phytocannabinoids”, of which over 100 are produced by the female *Cannabis* inflorescence [2]. The first evidence for the medical use of *Cannabis* dates to the Han Dynasty in ancient China, where it was recommended for pain, constipation, agitation, hysteria, spasmodic cough, disorders of the female reproductive tract, and other less defined conditions [3]. Of the 100 or so phytocannabinoids, the most potent is Δ-9-tetrahydrocannabinol (THC), which was isolated and identified as a major psychoactive compound in the 1960s [4]. This was followed by the discovery of additional phytocannabinoids, such as cannabidiol (CBD), cannabinol (CBN), cannabichromene (CBC), cannabigerol (CBG), tetrahydrocannabivirin (THCV), and Δ-8-THC [5].

In the early 1990s, two different G-protein-coupled receptors able to interact with phytocannabinoids were discovered in the central nervous system and the spleen; these receptors are now called type 1 and type 2 cannabinoid receptors (CB_1_ and CB_2_), respectively [6,7]. Their discovery was shortly followed by that of their ligands—two specific endogenous bioactive lipids, *N*-arachidonoylethanolamine (also known as anandamide, AEA) and 2-arachidonoylglycerol (2-AG) from animal tissues [8,9]. Later, the metabolic enzymes that regulate the production and degradation of these endogenous cannabinoids (endocannabinoids; eCBs) were discovered, followed by ancillary ligands, receptors, and transporters. These altogether represent the “endocannabinoid system (ECS)”, which is ubiquitously distributed in the body [4,10,11], including both the male and female reproductive tissues [12,13].

### 1.2. The Endocannabinoid System: A Multifaceted Network

Although the cannabinoid receptors were originally identified in the central nervous system (CNS for type 1 cannabinoid receptor—CB_1_), where they regulate the psychotropic effect of THC [14], and the spleen (type 2 cannabinoid receptor—CB_2_), where they have immunomodulatory functions [15], it is now clear that they are found throughout the human body [16,17]. The eCBs as ligands not only bind to CB_1_ and CB_2_, but also bind to and activate or inhibit the actions of the orphan G-protein-coupled receptors GPR55 [18] and GPR119 [19]; moreover, intracellularly, they bind to and activate the transient receptor potential (TRP) channels TRPV1, TRPV2, TRPV3, TRPV4, TRPA1, and TRPM8 [20], widely expressed in female reproductive tissues [21,22,23,24,25,26,27,28,29,30], and the nuclear peroxisome proliferator-activated receptor (PPAR) isotypes α, γ, and δ [31,32,33,34,35,36,37,38], through which they alter gene transcription [39].

One of the most studied eCBs is AEA. It is firstly synthesised through a cascade of enzymatic actions, involving cleavage of membrane phospholipid precursors by the specific action of a calcium-dependent *N*-acyltransferase (NAT), followed by the catalytic activity of a specific phospholipase D called *N*-acylphosphatidylethanolamine-specific phospholipase D (NAPE-PLD) [40]. Degradation of AEA is regulated mainly by the activity of fatty acid amide hydrolase (FAAH), and to a lesser extent by that of *N*-acylethanolamine acid amidase (NAAA) [17]. Both enzymes break down AEA (and other *N*-acylethanolamines) into arachidonic acid (for AEA) and ethanolamine. The other major eCB, 2-AG, is generated from hydrolysis of diacylglycerol by two specific *sn*-1 diacylglycerol-lipases (DAGL α and β), and is degraded by a specific monoacylglycerol lipase (MAGL) and, to a lesser extent and in a different way, by the α/β hydrolase domain (ABHD) proteins 2, 6, and 12 [17]. AEA and 2-AG can also be subjected to oxygenation by cyclooxygenase-2 (COX-2) and by different lipoxygenase (LOX) isozymes, as well as by several cytochrome P450 monooxygenases [4]. While additional endocannabinoids have been suggested, namely *N*-arachidonoyl dopamine (NADA), virodhamine, and noladin ether, their physiological role remains as yet unclear [4,17].

In common with other lipids in the body, eCBs are limited in their distribution around the body and within the aqueous intracellular cytosol by their hydrophobic properties. Intercellular transport of eCBs requires extracellular transporters, although these have not yet been identified. What is known is that when eCBs reach a cell, they cross the cell membrane via a facilitated diffusion process mediated by a putative, specific eCB membrane transporter (EMT), whose molecular identity has not yet been identified [41]. This could be because it does not even exist [42], leading others to speculate that extracellular eCBs bind to a membrane carrier protein located within caveolae lipid rafts, form vesicles, and enter cells through endocytosis [41]. When inside a cell, eCBs are transported by distinct carriers that drive them to different cellular compartments. Comprehensive reviews that summarise recent work on these transporters, such as fatty acid binding proteins (FABPs), heat shock protein 70 (HSP70), albumin, FAAH-1-like AEA transporter (FLAT-1), and sterol carrier protein 2 (SCP-2), can be found in the literature [41,43].

All of these elements and additional eCB-like compounds, such as *N*-palmitoylethanolamine (PEA), *N*-oleoylethanolamine (OEA), and *N*-stearoylethanolamine (SEA) (which exert a CB_1_-/CB_2_-independent “entourage effect” whereby FAAH preferentially catabolises the eCB-like compounds and potentiates the activity of other eCBs [44]), are part of the ECS orchestra involved in a growing number of physiological and pathological processes, including those occurring in the female reproductive tract.

### 1.3. ECS in Female Tissues and Reproductive Events

The main elements of the ECS are all expressed in human female reproductive tissues, such as the ovaries [45], Fallopian tubes (oviduct) [46], uterus [47], and placenta [48] (Figure 1). They have also been localised to areas of the hypothalamus responsible for producing hormones, which act through the hypothalamic–pituitary–gonadal (HPG) axis to control a great number of reproductive functions [38]. In the human ovaries, CB_1_ and CB_2_ have been shown to be expressed in the granulosa cells of primordial, primary, secondary, and tertiary follicles, as well as in theca cells of secondary and tertiary follicles (Figure 1), with the highest expression at the time of ovulation [45]. Additionally, both receptors are expressed in the corpus luteum and corpus albicans, even in the absence of pregnancy [45]. Moreover, FAAH has been shown to be present within theca cells, but NAPE-PLD appears only in the granulosa of secondary and tertiary follicles, the corpus luteum, and corpus albicans [45].

In the Fallopian tube, CB_1_ is expressed primarily in the smooth muscle cells and in surrounding blood vessels, with lower expression in the cytoplasm of epithelial cells lining the lumen of the tube [46]. In the endometrium, CB_1_ mRNA and protein levels increase in the secretory phase, probably under the influence of progesterone [50], while CB_2_ expression is minimal at the beginning of the cycle and increases markedly during the late proliferate phase of the menstrual cycle [47]. Interestingly, CB_1_ mRNA is only present at low levels in both the Fallopian tube and the endometrium of women with an ectopic pregnancy [46].

In addition, growing evidence suggests that the ECS is a part of the diverse mechanisms that regulate the complexity of events that occur in the early stages of pregnancy, and especially within the placenta [51]. The first trimester trophoblast contains transcripts for both CB_1_ and CB_2_ [52], and their expression is important for the continuation of normal pregnancy [48,53]. Additionally, increased CB_1_ expression might cause abnormal decidualisation, which might impair trophoblast invasion and thus be involved in the pathogenesis of preeclampsia [54] and miscarriages [55]. The data thus suggest that elevated eCBs are detrimental to continued pregnancy [56,57]. Furthermore, expression of FAAH is decreased in the first-trimester placenta, extra-villous trophoblast columns, villous cytotrophoblasts, syncytiotrophoblasts, and macrophages; tissues where increased FAAH expression could normally be acting to protect the growing embryo from the detrimental effects of AEA [52,58] and can result in elevated AEA levels in pregnancy that may result in miscarriage. Overall, the main components of the ECS are expressed and indeed regulate some of the functions within the female reproductive tract, as summarised in Table 1 and discussed in detail in the next section. Unsurprisingly, dysregulation of the ECS has been associated with some pathologies of the reproductive system, such as endometriosis, adenomyosis, leiomyoma, miscarriage, ectopic pregnancy, pre-eclampsia, and different types of gynaecological cancer (Table 1) [51].

### 1.4. The Endocannabinoid System in Relation to Normal Gynaecological Tissues

There have been several reviews [95] on this, and these are summarised in Table 1. All the components of the ECS are present and active in all parts of the female reproductive tract. Here, they play roles that include oocyte production [45,59,96,97,98], oviductal transport [98,99,100], and blastocyst maturity and implantation [101], as well as in preparing the endometrium for implantation [47,102,103,104,105]. When the ECS is dysfunctional or interfered with by, for example, cannabinoids [106,107], human fertility may be impaired (reviewed in [12,101,108,109]) and there may be associated reproductive-tissue-dependent pathologies, such as endometriosis, miscarriage, ectopic pregnancy, or pre-eclampsia [51,101,108,110,111,112]. Two recent reviews [12,38] on the ECS in the female reproductive tract summarise what is known on this topic, but crucially, these omit some important details on the main ECS components in gynaecological cancers, which we hope to address in this review (see Table 1). In this context, the ECS has been studied the most in the human ovary [45,59,60,61,63,64,65,113], cervix [80,84,85,86,87], and endometrium [47,50,55,68,69,70,72,73,74,75,81], the most common gynaecological cancers ([95,114,115]); however, other female cancers have not been studied, as shown in Figure 2. Although the presence and actions of the most commonly studied endogenous ligands (AEA, 2-AG, OEA, and PEA) in human reproductive tissues have been demonstrated, the presence and actions of others, such as SEA, virodhamine, stearamide, and monoolein [12,30,38,45,49,51,54,57,69,78,95,101,105,108,112,116,117,118,119,120,121,122,123] have not. Studies on receptor expression and function are few, and although there are some published studies on NAPE-PLD and FAAH expression and action in the female reproductive tract, many other (endo)cannabinoid metabolising enzymes have yet to be investigated, either in the normal female reproductive tract or in their related neoplasms (Figure 1 and Figure 2; Table 1).

### 1.5. An Overview of Gynaecological Cancers

Tumours of the female reproductive system are a diverse group of neoplasms that have different epidemiological, pathological, and clinical features and treatment options [95,124,125]. The malignant forms (Figure 2) constitute approximately one out of six cancers in women [114]. In the UK, gynaecological cancers are an important cause of morbidity and mortality, and are in the top 10 most commonly diagnosed female malignancies of the last decade [126]. Approximately 8500, 7100, and 3100 new cases of uterine, ovarian, and cervical cancers, respectively, were diagnosed in 2011 [126]. In 2012, there were 2000, 4300, and 920 deaths from uterine, ovarian, and cervical cancers, respectively [126]. These numbers are not confined only to countries with accurately collated data, such as the UK. Globally, for example, the number of patients diagnosed in 2018 with gynaecological cancers was relatively high (>295,000 ovarian, >382,000 uterine, >569,000 cervical, >17,500 vaginal, and >44,000 vulval) [115]. Mortality from these gynaecological cancers is high; for example, in 2018, approximately 185,000 women died from ovarian cancer, 90,000 from uterine cancer, 311,000 from cervical cancer, 15,000 from vulval cancer, and 8000 from vaginal cancers [115]. Unfortunately, similar data for Fallopian tube (oviductal) cancer and choriocarcinoma are unavailable, although there is increasing recognition that most surface epithelial ovarian cancers are of fimbrial origin. The global problem of increasing mortality from gynaecological cancers was recently highlighted by the World Health Organization as needing a response, and it thus pledged to eradicate cervical cancer by 2030 [127].

Significant progress has been made in reducing the incidence of some cancers, but the same cannot be said for some cancers of the female reproductive tract because a lack of a thorough understanding of their causes [128,129]. Cancers, including those of gynaecological origin, are distinguished by dysregulation of important cellular mechanisms, including those involved in the control of cell division, cellular differentiation, and apoptosis. The ECS is one of many factors thought to be involved in the development of cancers [130,131]. Interactions of this system with exogenous cannabinoids can potentially ameliorate [132] or exacerbate [133] the development or progression of cancer. We will now look at each of these interactions in some detail.

### 1.6. The Effects of Cannabinoids on Gynaecological Cancers

The main cannabinoids that are considered to have pharmaceutical promise in the treatment of cancer are the potent psychoactive and the commonly investigated non-psychoactive components of the *Cannabis* species, tetrahydrocannabinol (THC) and cannabidiol (CBD). Although there is scant evidence for their effectiveness in the treatment of gynaecological cancers, they are often promoted on medical cannabis production and distribution sites as having proven effectiveness [134,135,136,137,138,139]. Here, we examine the evidence in the scientific and clinical literature to support the current and future use of such compounds in the treatment of gynaecological cancers. These data are summarised in Table 1.

#### 1.6.1. Cannabinoids and Ovarian Cancer

Among gynaecological cancers, those of the ovary have the highest morbidity and mortality rates [140]. In an attempt to establish if there is a possible role for the ECS in ovarian pathophysiology, we [45] studied the expression levels of different components of the ECS [50], and demonstrated expression of CB_1_, CB_2_, and the NAE-modulating enzymes NAPE-PLD and FAAH in normal human ovaries using immunohistochemistry [45]. Additionally, AEA concentrations in follicular fluid after ovarian stimulation by hormones (following an in vitro fertilisation protocol that caused an increase in follicle size) were directly correlated with follicle size, suggesting that AEA is indeed involved in the hormonal maturation of follicles and oocytes [45,50]. Furthermore, data exist to indicate that AEA, OEA, and PEA are all elevated in follicular fluids of ovarian cancer patients and women with ovarian cysts [59].

Bagavandoss and colleagues demonstrated CB_1_ and FAAH expression in ovarian surface epithelium, the site from which some ovarian cancers often arise, providing another clue for a possible involvement of the ECS in ovarian cancer [96]. Regarding the expression of CB_1_ in ovarian cancer, Messalli and coworkers [141] showed that CB_1_ expression was moderate in benign and borderline epithelial rat ovarian tumours, but was increased in invasive ovarian tumours, suggesting a correlation between the extent of expression of the ECS components and the prognosis for patients with more aggressive ovarian cancer [141]. The levels of lysophospholipids such as lysophosphatidylinositol (an endogenous GPR55 agonist) in blood and ascitic fluids were also found to be elevated in ovarian cancer patients compared to healthy controls, a finding associated with proliferation and the metastatic potential of ovarian cancer cells [142]. Hofman and colleagues [143] more recently found that elevated lysophosphatidylinositol levels in the ovarian cancer cell lines OVCAR-3, OVCAR-5, and COV-362 resulted in GPR55-dependent angiogenesis. Their conclusion was based on experiments where pharmacological inhibition and genetic deletion of GPR55 reduced the pro-angiogenic potential of lysophosphatidylinositol in these cell lines. Additionally, they demonstrated that the mitogen-activated protein kinase pathway triggered via GPR55 by phosphorylation of ERK1/2 and p38, which are signalling molecules known to be involved in proliferative and migratory responses, could be curtailed by chemical interventions [143]. This observation suggests that some ovarian cancers might be amenable to pharmaceutical intercession. In addition, other components of the endocannabinoid system are important here. For example, the 2-AG degrading enzyme MAGL has been shown to be upregulated in aggressive human ovary cancer cells [65], and it is also thought to be involved in oncogenic signalling and, hence, in increased migration, invasion, and survival of many other cancer cell types [144]. These data suggest that identification of an effective drug that targets the ECS to treat ovarian cancer may have applications in the treatment of other cancers too. The application of such therapies would need to be timely, because MAGL overexpression in non-aggressive cancer cells often results in tumours that subsequently exhibit an increased pathogenic phenotype [65]. Moreover, the application of an MAGL inhibitor led to a reversion of the enhanced pathogenicity [65]. Thus, the involvement of the ECS, and especially the 2-AG signalling pathways in ovarian cancer, may fuel expectations on new therapeutics to combat this and other types of cancer. Some preliminary evidence suggests that OEA and its structural analogues may also have a beneficial effect on inhibiting ovarian cancer growth, but these data need to be confirmed in vivo [60]. There is little evidence that plant-derived (phyto)cannabinoids have any effect on ovarian biology or ovarian cancer development or progression, a concept that came from a study where SKOV-3-derived tumours were grown on the chorioallantoic membrane of fertilised chicken eggs [145], and then were treated with CBD-containing nanoparticles. The data indicated that CBD caused a 1.35- to 1.50-fold reduction in tumour size depending on the type of CBD formulation used [145]. The authors indicated that these nanoparticle preparations might be useful in the treatment of peritoneal metastases of ovarian cancer, possibly with lower adverse drug effects [145]. Furthermore, the preparations also reduced SKOV-3 ovarian cancer cell numbers in vitro, to almost zero within 48 h, possibly making this a good candidate for a randomised clinical trial. Of course, many additional studies are required before any candidate CBD formulation can be used in such clinical trials.

#### 1.6.2. Cannabinoids and Fallopian Tube Cancer

Fallopian tube cancer is a relatively rare gynaecological cancer (Figure 2). It is often categorised as being part of ovarian cancer (especially as there is emerging evidence that most surface epithelial ovarian cancers maybe of fimbrial origin), but it is important to study it as a separate entity. Just like other parts of the female reproductive tract, the oviduct (Fallopian tube) expresses all the components of the ECS, with CB_1_ and FAAH expression intimately associated with proper oviductal function [49,98,99]. When dysfunctional, the risk of ectopic pregnancy is markedly increased [46,49,99,100,146]. There is little evidence on the effect of cannabinoids on human oviductal function, but in the murine oviduct [100], THC reduces fertility because of the increased number of ectopically implanting embryos. In the bovine oviduct, there is gradation of AEA, OEA, and PEA concentrations in the oviductal epithelial cells with low levels in the isthmus and significantly higher levels of OEA and PEA (but not AEA) in the ampulla at the same point of the oestrous cycle [67]. These levels significantly fluctuated during the oestrous cycle [67], as they do in the human oviduct (Fallopian tube) during the menstrual cycle and along its length [49], with OEA causing a reduction in epithelial cell cilia beat frequencies [146], an effect that is likely to prevent timely movement of fertilised oocytes and precipitate ectopic pregnancy [49]. Although possible relationships between the ECS, cannabinoids, and oviductal cancer currently do not exist (Table 1), the fact that dysregulation of the ECS in the fallopian tube is related to the development of ectopic pregnancy makes us speculate that there could be a role for the ECS in oviductal cancer, and that such a possibility deserves to be investigated.

#### 1.6.3. Cannabinoids and Endometrial Cancer

Endometrial cancer, which is classified into type 1 and 2 [147], is the fourth most common cancer in women [148] and the most common gynaecological cancer. Various therapies exist depending on the disease grade and stage. Prognosis is poor, especially in those women with late presentation/detection [147]. Guida and coworkers [74] reported an upregulation of CB_2_ expression in endometrial cancer, whereby immunostaining was only successful in transformed malignant cells, while being completely absent in normal endometrial tissue. Furthermore, 2-AG levels were increased, but MAGL expression was decreased in comparison to controls, while AEA levels and FAAH expression were unaffected [74]. Similarly, Jove and colleagues [117] demonstrated that CB_1_ and CB_2_ were expressed at higher levels in stage III and IV endometrial carcinoma that has a poor prognosis. Unlike Guida and coworkers, the latter researchers found, by immunohistochemistry, an increase in CB_1_ expression, but no change in CB_2_ expression in stage 1 endometrial carcinoma tissue compared to normal endometrial tissue [117]. These observations were at odds with those of Risinger and coworkers, who found a decrease in CB_1_ receptor at the transcriptional level in stage 1 tissue [75]. These contradictory observations prompted us to investigate the ECS in endometrial cancer, using more than a single technique to interrogate CB_1_ and CB_2_ expression in endometrial cancer [70,72,73]. Our data indicated that CB_1_ and CB_2_ expression are decreased not only at the transcript level, but also at the protein level in both types 1 and 2 (stage 1) endometrial cancers (Table 1; Figure 3). We concluded that the discrepancy between these and previous studies was due to technical issues in the different methodologies used, including tissue sampling [72,73]. Furthermore, we examined the concentrations of plasma and tumour levels of AEA, OEA, and PEA in women with and without endometrial cancer, and showed that although the levels of all three *N*-acylethanolamines were increased in the tumours and in blood, only AEA and PEA were significantly higher in the plasma of such patients [29,68]. These data suggest that the differential catabolism of these three *N*-acylethanolamines might explain the different patterns of expression in endometrial cancer and plasma. We subsequently discovered that the apparent discrepancy between the tissue levels and plasma concentrations of OEA in the sample patient cohort was due to a decrease in the expression of FAAH in the tumour [78], without any change in the expression of NAPE-PLD (Table 1; Figure 3). The latter study also allowed us to define cut-off values for plasma AEA, OEA, and PEA concentrations (>1.36, >4.97, and 27.5 nM, respectively) that could be used in the prediction of endometrial cancer in symptomatic women [39], an observation that awaits confirmation in a larger, multicentre trial.

The effects of phytocannabinoids, such as THC, on the progression of endometrial cancer were recently evaluated by Zhang and collaborators [149]. They found that THC inhibited endometrial cancer cell proliferation and migration through decreased expression of matrix metalloproteinase-9, an effect mimicked by matrix metalloproteinase-9 gene silencing [149]. More recently, the effect of THC and CBD on endometrial cancer cell survival was investigated on Ishikawa and Hec50co cells [76], which are models of type 1 and type 2 endometrial cancer, respectively. The expression of all components of the ECS, including TRPV1, was demonstrated in these cells, supporting our in vivo observations (Figure 3). Additionally, treatment of the cells with AEA or CBD (>5 μM) reduced cell viability and was linked to an increase in reactive oxygen species production and caspase-3/-7 activity, which are markers of apoptosis [76]. Interestingly, in both endometrial cancer cell lines, THC had no effect on tumour cell survival, suggesting that in vitro findings in cancer cell lines cannot directly be translated to the in vivo situation. It is also interesting to note that the doses of CBD used in the in vitro study [76] greatly exceed those which are possible to achieve through recreational use of *C. sativa* or *C. indica*; this observation possibly explains the lack of anecdotal reports [136,138,139] that support anti-tumour benefits of marijuana use in patients with endometrial cancer [150]. The findings regarding the pro-apoptotic action of AEA in endometrial cancer cell lines are in keeping with the observations by Contassot and colleagues [84], who described AEA-driven cervical cancer cell apoptosis via TRPV1 activation. The danger from such studies, however, is that the data can be misinterpreted by agencies on the internet who wish to sell their *Cannabis* products, without an appreciation of the dangers they are potentially placing their customers in [138,139,150], and as summarised in [136], “…As always, the results must be met with scepticism and caution. The concentrations of cannabinoids used in these tests are quite high; it may not be safe to administer the amounts necessary to reach these concentrations. Furthermore, cells growing in a dish are very different from cancer cells in the body. They generate their own signals that cause them to grow out of control, which may counteract the effects of cannabinoids. In addition, individual genetic differences may influence how any particular patient will respond to medication. Nevertheless, these results point to CBD and other cannabinoids as a potential treatment for this common type of cancer…” [136]. This level of scepticism is appropriate because the drugs being promoted as anticancer therapies have not undergone the rigorous pre-clinical studies and randomised clinical trials for the treatment they are being advertised for [136,138,139,150]. The danger here is that the multiple pleiotropic effects that cannabinoids (especially THC and CBD) exert on the female body have not been discovered, and adverse side-effects of self-treatment by patients with gynaecological cancers, either by topical or oral routes, could result in serious morbidities or mortality, especially as the effects of these phytocannabinoids go well beyond the effect of apoptosis [151]. Indeed, interactions between (endo)cannabinoids and the stromal cells of the endometrium are often ignored in cancer studies, and these cells of the endometrium are also affected by (endo)cannabinoids [55,71]. It is thus essential that more laboratory-based and clinical studies in this area are performed.

#### 1.6.4. Cannabinoids and Cervical Cancer

Cervical cancer is the second leading cause of malignancy-related deaths in women worldwide due to the lack of customisable and effective treatments (especially in low- and middle-income countries), with more than 250,000 deaths being reported annually [152]. A possible role of the ECS in the development of cervical cancer has been elucidated in recent years. Contassot and coworkers [84] reported a strong expression pattern of CB_1_ and CB_2_, as well as of TRPV1, in cervical carcinoma cell lines and biopsies. In addition, it was shown that AEA had a pro-apoptotic effect on cervical carcinoma cell lines (HeLa and Caski) [84], which were not inhibited, but were instead enhanced by CB_1_ and CB_2_ antagonists. On the other hand, the TRPV1 selective antagonist capsazepine protected the cell lines from AEA-induced apoptosis, indicating an important role of the TRPV1 channel in the pro-apoptotic action of AEA [92]. Additionally, it was demonstrated by Ramer and collaborators [153] that CBD decreased the invasiveness of cancer cells in a concentration-dependent manner. This effect was observed in the cervical cancer cell lines HeLa and C33A, as well as in the lung cancer cell line A549, and seemed to be mediated by the upregulation of TIMP-1 via CB_1_/CB_2_ and TRPV1. TIMP-1 is an inhibitor of matrix metalloproteinases, and as such, it prevents the movement of cells out of the tissue and, hence, a metastatic disease, as has been observed in a patient with ovarian cancer treated with CBD [154].

The activation of p38 and p42/44 mitogen-activated protein kinases was identified as an upstream event in TIMP-1 upregulation [153]. In agreement with these findings, it was reported that treatment of different cervical cancer cell lines (HeLa, SiHa, ME-180) with CBD led to a decrease in cell proliferation [155]. Furthermore, CBD induced cell death by the accumulation of cells in the sub-G0 phase (cell death phase) of the cell cycle, a finding that was most likely caspase-dependent because caspase-9 as well as caspase-3 were upregulated upon CBD treatment [155]. Hence, CBD may be an additional therapeutic tool for the treatment of cervical cancer, yet additional in vivo studies, similar to that performed on a single ovarian cancer patient [154], will be needed to clarify the impact of CBD on cervical cancer.

#### 1.6.5. Cannabinoids and Vaginal Cancer

Vaginal cancer is uncommon, and the American Cancer Society estimated that >6000 women will be diagnosed with it in 2020. The estimated lifetime risk is 1 in 1100 (i.e., less than 0.1%). Of the 6000 USA women expected to be diagnosed with vaginal cancer in 2020, 1450 will die because they have this disease [156]. The role of the ECS in vaginal cancer has not been fully examined. We [80] have demonstrated that CB_1_ and FAAH are expressed in the normal vagina; however, there are no data on the expression of other components of the ECS (Table 1), nor on what their normal function might be. What happens to the expression of these factors or what effects cannabinoid and eCB ligands might have on the vagina or on cells of vaginal tumours is uncertain/unclear (Table 1). The internet is one source of information, and for the vagina, it is reported that some women experience a “vaginal high” when using cannabinoids, especially as a topical application [135]. The problem with these data is that only 40% of women experience this “psychological” effect [135]. Nevertheless, these statements have led some internet sites to suggest that different *cannabis*-containing preparations might be useful for the treatment of some of the symptoms associated with vaginal cancer [134,157]. Obviously, a lot more information is needed on the role of cannabinoids and eCBs in the human vagina, and especially in vaginal cancer.

#### 1.6.6. Cannabinoids and Vulvar Cancer

Vulvar cancer is a less common gynaecological cancer [158]. The vulva is very similar to normal thin skin and is known to express CB_1_ and FAAH [80], but it is not known if it contains all the main components of the ECS (Table 1). The only existing evidence that cannabinoids have an effect on the vulva comes from a less-than-reliable internet source [137]. A *C. sativa* ethanolic extract and a purified CBD preparation had anti-inflammatory effects on keratinocytes and skin fibroblasts in vitro, suggesting that CBD was the main active ingredient that would be effective in wound injury [159].

This seems important because women with vulvar cancer often undergo radical surgery to remove their malignancy, which causes disfiguration of the female external genitalia, and causes significant long-term emotional and physical instability [160]. Indeed, the use of the CBD derivative VCE-004.3 on skin fibrosis and inflammation [161] demonstrated a CB_2_/PPARγ-dependent effect, and suggested that similar compounds might be beneficial for patients with vulvar cancer who have undergone surgery and need topical treatment for the pruritus; the latter is associated with skin fibrosis and inflammation, especially as VCE-004.3 appears to inhibit mast cell degranulation [161]. The toxicity profile of such topical administrations remains to be determined; however, ethanolic extracts of THC, CBD, and other cannabinoids appear in the blood shortly after administration; thus, some caution is advised, also in the light of the pleiotropic effects of these compounds [162]. Obviously, more detailed analysis of the role of the ECS and of plant-derived cannabinoids in the treatment of vulvar cancer is warranted.

#### 1.6.7. Cannabinoids and Choriocarcinoma

The function of the female reproductive tract is to support the embryo and fetus during its development into an independent offspring (Figure 1). In order to do this, the coordinated actions of many interacting factors need to take place, of which the ECS is an integral part [101,108,111,163,164,165,166]. A key tissue in human reproduction is the fetoplacental unit. The entire ECS is present in the placenta [166] (see also Table 1), and modifications of its components result in obstetrical problems, such as miscarriage [56,57,113,167], babies that are small for gestational age [168,169], and pre-eclampsia [170]. In addition, dysregulated *N*-acylethanolamine levels may be responsible for preterm delivery [121,123]. The placenta can also undergo neoplastic changes into two clinically relevant conditions, hydatiform mole (a non-malignant transformation) and choriocarcinoma (a malignant transformation), which appear noteworthy. Currently, there are no data on the expression of the ECS in either of these tumours; there is, however, evidence that AEA and THC both affect a model for choriocarcinoma, like BeWo cells [52,93,94,171], and a model for normal trophoblast, like TCL-1 cells [172], where cell growth is affected mainly through a CB_2_-dependent mechanism [94,171,172]. These observations, coupled with evidence that THC decreases STAT3 signalling in mice with reduced fetus numbers and placental weights [168], support the view that cannabinoid use in human pregnancy is likely to affect the placenta in a similarly dangerous manner [173]. The increased use of CBD in pregnancy as an anti-emetic [174,175] is thus of great concern because the toxicity profile of CBD in pregnancy is not fully known [174,176,177], and especially as CBD can inactivate both placental CB_1_ and CB_2_ receptors in vitro [178].

## 2. Conclusions

A pivotal role of the ECS in gynaecological cancers has been demonstrated in recent years; in particular, the development, progression, and prognosis of female reproductive tract diseases seem to be associated with their dysregulation [12,30,38,56,57,95,105,108,112,113,118,119,120,122,141,179]. Due to manifold cellular and metabolic regulatory functions, the ECS represents an important therapeutic target that needs further investigation. Cannabinoids, especially plant-derived or synthetic compounds that impact eCB signalling as specific agonists or antagonists of their receptor targets, may potentially influence the functional dysregulation that is apparent in gynaecological cancers. For this reason, more research is required to shed light on the complex interactions of the ECS with respect to the administration of preparations derived from *C. sativa* or *C. indica* in order to find new therapeutic tools for effective and safe therapy of gynaecological cancers. One of the main limitations of available studies is that endogenous and exogenous cannabinoids behave differently, and their modes of action in vivo and in vitro are difficult to correlate. It has also become apparent that the effects of cannabinoids vary in a dose-dependent manner. It is thus important to keep these factors in mind when trying to reconcile inconsistent results between studies. Large systematic reviews and meta-analyses would be helpful to sift through these studies, their methods, and their results in order to reach conclusions about treatment efficacy. A recent large systematic review by Whiting and colleagues [180] concluded that the use of cannabinoids in the treatment of a variety of conditions, such as multiple sclerosis, glaucoma, and chronic pain, was associated with adverse effects, such as disorientation, gastrointestinal upset, emesis, and fatigue [180]. Unfortunately, this review did not address cannabinoid use in female reproductive conditions, nor did it supply data about adverse effects of topical cannabinoid administration (e.g., for vulvar disease [134] or the less common vaginal highs [135]). While the political and social environment is becoming more tolerant of medical cannabinoids, the stigma surrounding *cannabis* use and its derivatives still represents a barrier to effective clinical research [181]. More recently, this attitude has appeared to be changing, yet there is a dearth of available health-authority-regulated cannabinoid compounds [181], and, as such, many patients may look to the internet to find unregulated and untested medicinal products [134,135,136,137,138,139]. Without rigorous regulation and testing of such compounds [182], there is no way of knowing exactly what these products contain and whether they could be harmful to patients. Nevertheless, increasing research into cannabinoid treatments could potentially expand the number and variety of therapies available to cancer patients and limit the need for unregulated products.

In summary, the ECS in the female reproductive tract is fully functional, but intricate in its interactions. Current knowledge on all the components of this system in the reproductive tract is incomplete, and thus, a full picture remains elusive. Although it is composed of multiple receptors, the female reproductive tract is stimulated by numerous exogenous cannabinoids and eCBs, and multiple metabolic enzymes that regulate eCB levels and activity are known; knowledge on the various roles that each of these components have on the initiation, development, and progression of benign and malignant tumours of the female reproductive tract is lacking, or, at best, at an early stage. Much is still unknown (Table 1), and although many studies over the past two decades have highlighted the critical role of the ECS in maintaining key aspects of human and animal reproduction, including immune modulation, inflammation, cell proliferation, and differentiation [101,165,183], the precise roles of these factors in common reproductive tract cancers remain poorly defined. The roles, if any, in less common reproductive cancers (such as those of the vagina, vulva, and trophoblast/placenta) are untested and could provide fertile ground for subsequent studies. Further investigations into the specific influences of cannabinoid type, receptor affected, delivery method, chemical composition, and component concentration [184,185] will help to elucidate the intricacies of the role that the ECS plays in gynaecological cancers. Doing so will provide an excellent opportunity to expand the therapeutic arsenal for treating female neoplastic diseases.

## Figures and Tables

**Figure 1 cancers-13-00037-f001:**
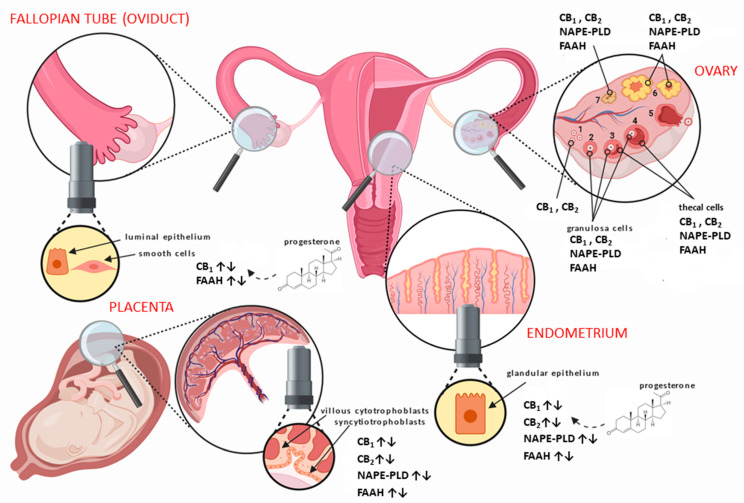
Distribution of the main endocannabinoid system (ECS) components in human female reproductive tissues. In the ovary, the different stages of follicular development from (1) primordial/primary, (2) secondary, (3) tertiary, (4) pre-ovulatory/Graafian, to (5) ovulating follicles are depicted. After ovulation is complete, the condensing granulosa and mural thecal cells form the corpus luteum (6), a structure that produces the progesterone required for continued early pregnancy. In the absence of pregnancy, the corpus luteum degenerates into the corpus albicans (7). Throughout the ovarian cycle, CB_1_ (type 1 cannabinoid receptor) and CB_2_ (type 2 cannabinoid receptor), fatty acid amide hydrolase (FAAH), and *N*-acylphosphatidylethanolamine-specific phospholipase D (NAPE-PLD) are produced in the various cells of the developing follicle and corpus luteum, including the oocyte [45]. Similarly, CB_1_, CB_2_, FAAH, and NAPE-PLD are expressed in the Fallopian tube [49] and endometrium [47] throughout the menstrual cycle, where they are regulated by the actions of estradiol and progesterone. The cytotrophoblast and syncytiotrophoblast cells of the early placenta also express CB_1_, CB_2_, FAAH, and NAPE-PLD [48], where modulation of protein expression occurs when production of progesterone changes from the corpus luteum to the placenta.

**Figure 2 cancers-13-00037-f002:**
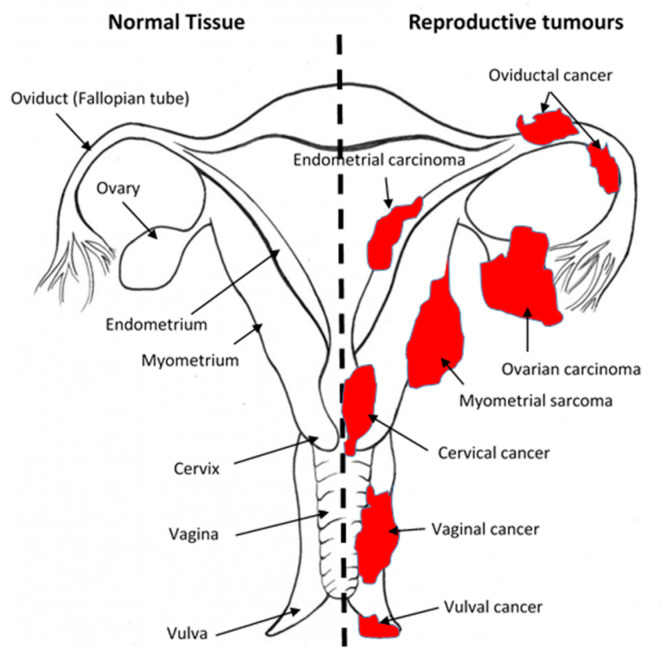
Sites of gynaecological cancers in the female reproductive tract. The diagram indicates the names of the normal tissues of the female reproductive tract (left side) and the sites and names of the cancers (right side) for the corresponding normal tissues. Please add copyright if necessary.

**Figure 3 cancers-13-00037-f003:**
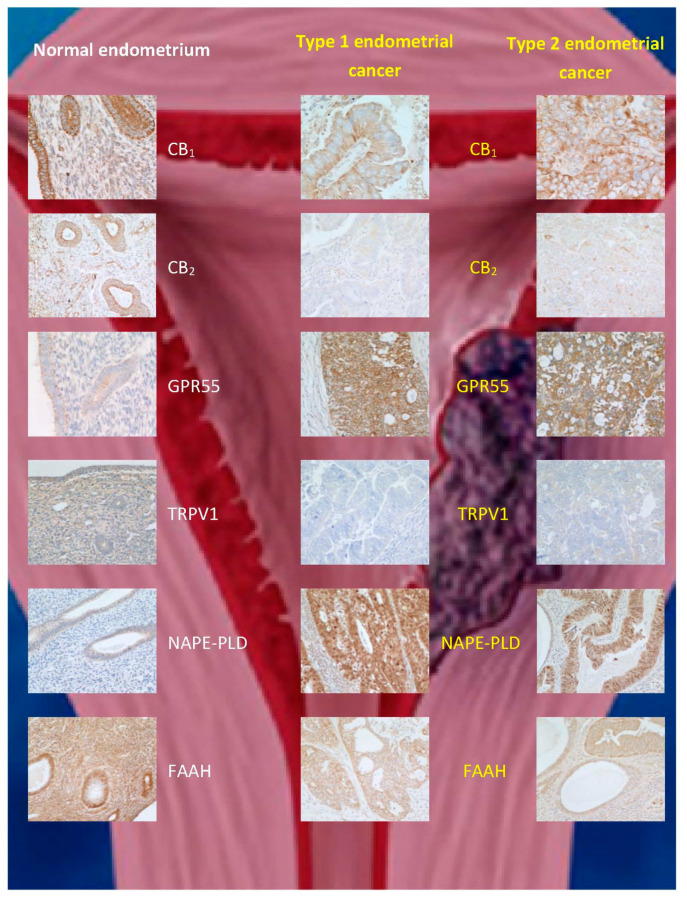
Immunohistochemical staining patterns for ECS proteins in normal endometrium and (type 1 and type 2) endometrial cancer. The data are taken from [74], where validation for the commercial antibodies and techniques used can be found. Note the reduction in CB1, CB2, TRPV1, and FAAH protein staining and increases for GPR55 and *N*-acylphosphatidylethanolamine-specific phospholipase D (NAPE-PLD) expression in both types of endometrial cancer when compared to that of normal tissue.

**Table 1 cancers-13-00037-t001:** Summary of the main ECS components in the female reproductive tract.

Tissue	ECS Component	Normal Tissue	Change in Cancer	Pertinent References
Ovary	AEA	yes	↑	[45,59]
	OEA	yes	↑	[59,60]
	PEA	yes	↑	[59]
	2-AG	yes	↑	[61]
	CB_1_	yes	?	[45]
	CB_2_	yes	?	[45,62]
	TRPV1	yes	↑	[63]
	GPR55	no	↑	[64]
	NAPE-PLD	yes	?	[45]
	FAAH	yes	?	[45]
	DAGL	?	?	none
	MAGL	?	*↑	[65]
Fallopian Tube (Oviduct)	AEA	yes	?	[59]
	OEA	yes	?	[59]
	PEA	yes	?	[59]
	2-AG	?	?	none
	CB_1_	yes	?	[46,49,66]
	CB_2_	yes	?	[49,66]
	TRPV1	?	?	none
	GPR55	?	?	none
	NAPE-PLD	yes	?	[49]
	FAAH	yes	?	[49,67]
	DAGL	?	?	none
	MAGL	?	?	none
Uterus (endometrium)	AEA	yes	↑, ±	[50,55,68,69]
	OEA	yes	↑	[68]
	PEA	yes	↑	[68]
	2-AG	yes	↑	[12]
	CB_1_	yes	↓, ±, ↑	[47,70,71,72,73,74,75]
	CB_2_	yes	↑, ↓	[47,72,74]
	TRPV1	yes	↓	[76,77]
	GPR55	yes	↑	[77]
	NAPE-PLD	yes	↑	[47,78]
	FAAH	yes	↓	[47,78,79,80]
	DAGL	yes	?	[74,81]
	MAGL	yes	?	[74,81]
Uterus (myometrium) **	AEA	Yes	?	[30]
	OEA	Yes	?	[30]
	PEA	Yes	?	[30]
	2-AG	?	?	none
	CB_1_	Yes	?	[30,82,83]
	CB_2_	Yes	?	[30,83]
	TRPV1	Yes	?	[30]
	GPR55	yes	?	[30]
	NAPE-PLD	yes	?	[30,82]
	FAAH	yes	?	[30,82]
	DAGL	yes	?	[81]
	MAGL	yes	?	[81]
Cervix	AEA	yes	?	[84,85]
	OEA	?	?	none
	PEA	?	?	none
	2-AG	?	?	none
	CB_1_	yes	?	[80,84]
	CB_2_	yes	?	[80,84,86]
	TRPV1	yes	?	[84,85,87]
	GPR55	?	?	none
	NAPE-PLD	?	?	none
	FAAH	yes	?	[80]
	DAGL	?	?	none
	MAGL	?	?	none
Vagina	AEA	?	?	none
	OEA	?	?	none
	PEA	?	?	none
	2-AG	?	?	none
	CB_1_	yes	?	[80]
	CB_2_	?	?	none
	TRPV1	?	?	none
	GPR55	?	?	none
	NAPE-PLD	?	?	none
	FAAH	yes	?	[80]
	DAGL	?	?	none
	MAGL	?	?	none
Vulva	AEA	?	?	none
	OEA	?	?	none
	PEA	?	?	none
	2-AG	?	?	none
	CB_1_	yes	?	[80,88]
	CB_2_	yes	?	[88]
	TRPV1	yes	?	[88]
	GPR55	yes	?	[88]
	NAPE-PLD	?	?	none
	FAAH	yes	?	[80]
	DAGL	?	?	none
	MAGL	?	?	none
Placenta (Trophoblast) ***	AEA	yes	?	[53,89]
	OEA	yes	?	[53]
	PEA	yes	?	[53]
	2-AG	yes	?	[90]
	CB_1_	yes	?	[48,53,58,80,91]
	CB_2_	yes	?	[48,53,58]
	TRPV1	yes	?	[48]
	GPR55	yes	?	[92]
	NAPE-PLD	yes	?	[48,53]
	FAAH	yes	?	[48,53,58,80,91]
	DAGL	yes	?	[93,94]
	MAGL	yes	?	[93,94]

Notes: Yes = present; No = absent; ? = currently unknown; ↑ = increases; ↓ = decreases; ± = unchanged; none = no work in this area; *↑ in aggressive tumours when compared to levels in non-aggressive tumours; ** malignant tumour known as uterine sarcoma and non-malignant precursor known as leiomyoma; *** malignant tumour known as choriocarcinoma and non-malignant precursor known as hydatiform mole.

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
