# Peer review of "(Endo)Cannabinoids and Gynaecological Cancers"

_cancers, 2020, doi:10.3390/cancers13010037_

Round 1
Reviewer 1 Report
The review is interesting and the concepts and the idea in which is based are of great interest nowadays, although I think it need a major restructuration in the following points:
- The abstract, specially from line 19 to line 23 talks about very general concepts. The abstract should describe the article in a more precise and detailed manner.
- The article must improve its structure. The article is structured in two general points: point 1 corresponds to the introduction with different subsections and point 2 which is directly the conclusions. In my opinion, the structure could be: introduction (formed by 1.1 and 1.2), a whole new section for 1.3 and 1.4 (ECS in female tissues and reproductive events), another new section for ECS in gynaecological cancers and the last one with the conclusions.
- The text mentions several times that the authors have made different experiments employing the expression “we reported”. I think the article must be rewrite using the third person.
- Lines 49-50: Indicates that CBD is less studied than THC, although CDB has been extensively evaluated in the medicine field, even with commercial formulations.
- Review the expression of the CB2 receptor in the immune system
- Figure 1 legend: It’s a extensive legend which can be incorporated to the text for a better explaining. Also the image corresponding to the placenta should be better specified.
- Lines 188-190: data on incidence and mortality of the different types of cancer should be more actual and more global (not only from UK). Moreover, data should be more consistent (in the description of some types of cancer there’s a lack of data on incidence or mortality).
- Line 202: Improve the explanation between CBD and bio-oil because the bio-oil could also be a set of cannabinoids.
- Line 261: Controversial affirmation because is based on results from in vitro and in ovo studies. It could be better if these results were confirmed in vivo.
- Lines 386-392 and line 397: I don’t think is appropiate to use internet in general terms as the source in an article of this level. I suggest rewrite or remove both parts or search for a better source.
Author Response
Report 1:
The review is interesting and the concepts and the idea in which is based are of great interest nowadays, although I think it need a major restructuration in the following points:
- The abstract, specially from line 19 to line 23 talks about very general concepts. The abstract should describe the article in a more precise and detailed manner.
Response: We have moved the very general concepts of the original abstract into the ‘lay abstract’. Now, a more detailed abstract has been provided in the revised manuscript, as kindly suggested.
- The article must improve its structure. The article is structured in two general points: point 1 corresponds to the introduction with different subsections and point 2 which is directly the conclusions. In my opinion, the structure could be: introduction (formed by 1.1 and 1.2), a whole new section for 1.3 and 1.4 (ECS in female tissues and reproductive events), another new section for ECS in gynaecological cancers and the last one with the conclusions.
Response: When we submitted this manuscript, no labelling for the section headings were present and these we applied by the Journal’s copy editors. We also noted that Reviewer #2 also wanted a different numbering system, and so we have used that numbering system instead. Of course, the Editors however may restructure the heading number as they feel is appropriate for Journal style and consistency.
- The text mentions several times that the authors have made different experiments employing the expression “we reported”. I think the article must be rewrite using the third person.
Response: Although we only used this phrase once in the article (original line 212) and wanted to take responsibility for those studies and the conclusions made, we accede this point and have used 3rd person language instead. The sentence now reads, “Additionally, AEA concentrations in follicular fluid after ovarian stimulation by hormones (following an in vitro fertilisation protocol that caused an increase in follicle size) were directly correlated, suggesting that AEA is indeed involved in the hormonal maturation of follicles and oocytes [42,46].”.
- Lines 49-50: Indicates that CBD is less studied than THC, although CDB has been extensively evaluated in the medicine field, even with commercial formulations.
Response: We have removed the phrase “less studied”.
- Review the expression of the CB2 receptor in the immune system
Response: Franckly, we believe that reviewing the CB2 receptor in the immune system is well beyond the scope of this article, and it could even distract the reader from the topic we aim at covering. Thus, we chose to include two comprehensive reviews on the subject, quoted as new references 15 and 16.
Figure 1 legend: It’s a extensive legend which can be incorporated to the text for a better explaining. Also the image corresponding to the placenta should be better specified.
Response: We have now produced a figure legend that we believe should be suitable for a novice to the female reproductive tract to understand the tissues and structures that are under scrutiny. Removal of the figure legend components and placing them in a revised part of the manuscript would significantly increase the amount of material the reader would need to read, in order to simply understand the anatomy of the human female reproductive tract. We have carefully examined the wording used and deemed the current length the minimum required to describe the contents of the figure. We do, however, agree that labelling of those structure, especially the placenta, could be improved, therefore we have revised the figure and added some additional labelling.
- Lines 188-190: data on incidence and mortality of the different types of cancer should be more actual and more global (not only from UK). Moreover, data should be more consistent (in the description of some types of cancer there’s a lack of data on incidence or mortality).
Response: The reason for using the UK as an example on the incidence and mortalities of the most common gynaecological cancers is that our Office for National Statistics accurately records all-cause mortality as part of a government strategy, where to place its limited healthcare resources on. Many other countries (including the USA, China and Russia) do not do this but provide only estimates and, in fact, only 1 in 5 countries provide such data. Despite its best efforts, the World Health Organization (WHO) recognises this limitation and provides best estimates on actual recorded incidences and deaths, not annually but only every few years. The last estimate provided was in 2018, and so we have provided global estimated values for each of the different types of cancer (where available) in lines 190 to 200, as requested. We have also provided a reference to a recent publication by the WHO on this issue and refer the Reviewer to the recent news item and conference on a global effort to eradicate cervical cancer that occurred on World Cancer Day (see Spotlight on cervical cancer | World Cancer Day).
- Line 202: Improve the explanation between CBD and bio-oil because the bio-oil could also be a set of cannabinoids.
Response: In line with the comment from Reviewer #3, we have removed all references to CBD as a bio-oil. We thank both Referees for bringing this relevant point to our attention.
- Line 261: Controversial affirmation because is based on results from in vitro and in ovo studies. It could be better if these results were confirmed in vivo.
Response: We agree with the Referee. Accordingly, we have added a qualifying sentence at the end of this section. It reads “Of course, many additional studies are required before any candidate CBD formulation can be used in such clinical trials.”
- Lines 386-392 and line 397: I don’t think is appropiate to use internet in general terms as the source in an article of this level. I suggest rewrite or remove both parts or search for a better source.
Response: This is a key point and secondary focus of the review article. Please note that Reviewer #2 identified this as being a key point that should be retained for this review articles, so we have retained it.

Reviewer 2 Report
In general, I liked the work presented by the authors. The review offers an overview, not only of the components of the endocannabinoid system and its (dys)regulation in different gynaecological cancers, but also of current studies and overview of possible therapies based on cannabinoids. Moreover, the authors discuss about the dangers of the misinterpretation based on the “information” found in general media (such as the internet) and the necessity of proper studies with cannabinoids for treating these pathologies determined give its potential. They not only discuss about the most frequent gynaecological cancers, but also about others not so known, which is a good point. The Table 1 is a very good summary to see at a glance what is known about ECS in those cancers and the necessity of more research in the field.
However, I have some suggestions for the authors:
- Line 20 “the products of conception; or secondary, originating from other parts of the body” : I would rather use metastasis (if this is what you mean with “from other parts of the body”), because it is not really clear the meaning in this form.
- Line 62 "2. The endocannabinoid system: a multifaceted network": In this section, in my opinion, there is too much explanation of the AEA biochemistry (transport, signalling, etc.) and I do not think it is necessary for the purpose of the review. I noticed that most of this information it is no so used in the following sections, although it highlights the complexity of the system. I would stick to a briefer explanation of the endocannabinoid system and its complexity, not so centered on the AEA.
- From the section 1.7 to 1.13, I would number them as 1.6.1 to 1.6.7 because I consider them as part of the section “1.6. The effects of cannabinoids on gynaecological cancers”. As well, when talking about the different cancers, I would place in the firsts positions the most frequents cancers and finish with the least frequents, i.e. cannabinoids and ovarian cancer, cannabinoids and endometrial cancer, cannabinoids and cervical cancer, cannabinoids and fallopian tube cancer, cannabinoids and vaginal cancer, cannabinoids and vulvar cancer, cannabinoids and choriocarcinoma cancer (the last four, as they are all no so frequent, it does not matter the order).
- Line 325 “vivo situation. It is also interesting to note that the doses used in the in vitro study [73] greatly exceed”: Maybe it should be indicated the cannabinoids doses that this is referring to (CBD or THC?) because with the context it is not clear.
Author Response
Report 2:
In general, I liked the work presented by the authors. The review offers an overview, not only of the components of the endocannabinoid system and its (dys)regulation in different gynaecological cancers, but also of current studies and overview of possible therapies based on cannabinoids. Moreover, the authors discuss about the dangers of the misinterpretation based on the “information” found in general media (such as the internet) and the necessity of proper studies with cannabinoids for treating these pathologies determined give its potential. They not only discuss about the most frequent gynaecological cancers, but also about others not so known, which is a good point. The Table 1 is a very good summary to see at a glance what is known about ECS in those cancers and the necessity of more research in the field.
Response: We thank you very much indeed for the kind words, and for your understanding of the goal and aim of the review article we were trying to produce.
However, I have some suggestions for the authors:
- Line 20 “the products of conception; or secondary, originating from other parts of the body” : I would rather use metastasis (if this is what you mean with “from other parts of the body”), because it is not really clear the meaning in this form.
Response: We have amended this sentence to convey the idea that secondary neoplasms in the form of metastatic disease was what we intended here. It now reads, “originating either from the reproductive tract or the products of conception; or secondary neoplasms, representative of metastatic disease.”
- Line 62 "2. The endocannabinoid system: a multifaceted network": In this section, in my opinion, there is too much explanation of the AEA biochemistry (transport, signalling, etc.) and I do not think it is necessary for the purpose of the review. I noticed that most of this information it is no so used in the following sections, although it highlights the complexity of the system. I would stick to a briefer explanation of the endocannabinoid system and its complexity, not so centered on the AEA.
Response: We have amended this section to include other eCBs whose physiological roles remain uncertain (lines 84 to 89) and included references to review articles that highlight that uncertainty. Additionally, we have edited lines 90 to 105 to indicate that the transport mechanisms that AEA uses to travel around the circulatory system, enter a cell and find its cellular targets are shared by the other eCBs and so are not AEA-specific.
- From the section 1.7 to 1.13, I would number them as 1.6.1 to 1.6.7 because I consider them as part of the section “1.6. The effects of cannabinoids on gynaecological cancers”. As well, when talking about the different cancers, I would place in the firsts positions the most frequents cancers and finish with the least frequents, i.e. cannabinoids and ovarian cancer, cannabinoids and endometrial cancer, cannabinoids and cervical cancer, cannabinoids and fallopian tube cancer, cannabinoids and vaginal cancer, cannabinoids and vulvar cancer, cannabinoids and choriocarcinoma cancer (the last four, as they are all no so frequent, it does not matter the order).
Response: This manipulation to the numbering of the section headings was performed by the Journal editors in line with Journal style. We have now amended numbering as requested, and hope the Editors agree that this manipulation would be an improvement. Please let us mention that we have dealt with each tissue in an anatomical hierarchy with the order from the ovary through the female reproductive tract from that primary regulator of the female reproductive tract to the exit point of the ova if it is not fertilised. If it is fertilised, then the pregnant uterus comes into play and so this becomes an additional section for inclusion in this review. This structure thus leads the reader from the starting point of human reproductive potential, the ovary, to the end point of the oocyte, the vulva.
- Line 325 “vivo situation. It is also interesting to note that the doses used in the in vitro study [73] greatly exceed”: Maybe it should be indicated the cannabinoids doses that this is referring to (CBD or THC?) because with the context it is not clear.
Response: We have clarified that it was CBD: “It is also interesting to note that the doses of CBD used in the in vitro study [73] greatly exceed those…”.

Reviewer 3 Report
This is well written review article that focuses on endocannabinoids in the female reproductive system with an additional focus on cannabinoid control of gynaecological cancers. The author also developed and informative review table and engaging figures.
- Others may argue there are four species: indica, C. sativa, Cannabis afghanica, and Cannabis ruderalis. See https://www.ncbi.nlm.nih.gov/pmc/articles/PMC5576603/. It may be best to focus just on Cannabis Sativa (Cannabis).
- Line 161, included vs updated.
- Line 174, however – add a comma after.
- Line 190, after respectively there are two periods.
- CBD, a bio-oil. Would recommend removing bio-oil.
- Line 236, the jump between the GPR55 study and the MAGL studies and therapeutic interventions was confusing. The author should clarify this section.
- Line 259, it is a significant jump to translate experiments from an ovarian cancer cell line in culture to suggesting clinical trials. The author could tone down this statement.
- Line 323, “Interestingly, in both endometrial cancer cell lines, THC had no effect on tumour cell..” This statement should be removed unless there is a reference for THC antitumor activity in vivo in endometrial cancer.
- Line 392, significantly vs a lot more.
- Line 444, the Author can remove all the references if they are all presented in the main body of the review.
- 12. Line 464, would replace marijuana with Cannabis.
Author Response
Report 3:
This is well written review article that focuses on endocannabinoids in the female reproductive system with an additional focus on cannabinoid control of gynaecological cancers. The author also developed and informative review table and engaging figures.
- Others may argue there are four species: indica, C. sativa, Cannabis afghanica, and Cannabis ruderalis. See https://www.ncbi.nlm.nih.gov/pmc/articles/PMC5576603/. It may be best to focus just on Cannabis Sativa (Cannabis).
Response: Some would also argue that there is only one species (as the quoted article also alludes to), but since pharmaceutical chemists have identified C. sativa and C. indica as being of chemical interest in their pursuit of the right combination of phytocannabinoids for animal and human therapy, we have taken the liberty to only list the 3 ‘major’ species of cannabis plant into the narrative. This small change has been incorporated into the main introduction (line 40).
- Line 161, included vs updated.
Response: we have changed the word ‘updated’ to ‘included’.
- Line 174, however – add a comma after.
Response: A comma has been added.
- Line 190, after respectively there are two periods.
Response: Thank you. The second period has been removed.
- CBD, a bio-oil. Would recommend removing bio-oil.
Response: We have removed ‘bio-oil’, also in keeping with Referee #1 point 7.
- Line 236, the jump between the GPR55 study and the MAGL studies and therapeutic interventions was confusing. The author should clarify this section.
Response: We have subdivided the ideas contained in this section to make our meaning clearer to the reader. The section now reads, ”This observation suggests that some ovarian cancers might be amenable to pharmaceutical intercession. In addition, other components of the endocannabinoid system are important here. For example, the 2-AG degrading enzyme MAGL has been shown to be upregulated in aggressive human ovary cancer cells [62] and it also seems to be involved in oncogenic signalling and hence in increased migration, invasion, and survival of many other cancer cell types [140]. These data suggest that identification of an effective drug to treat ovarian cancer may have implications for the treatment of other cancers too.”
- Line 259, it is a significant jump to translate experiments from an ovarian cancer cell line in culture to suggesting clinical trials. The author could tone down this statement.
Response: Please note that this was not a statement but a question. We have added an additional sentence at the end of this paragraph to qualify the ‘statement’ so as to clarify what the problem is for the reader. The added sentence reads, “Of course, many additional studies are required before any candidate CBD formulation can be used in such clinical trials.”
- Line 323, “Interestingly, in both endometrial cancer cell lines, THC had no effect on tumour cell.” This statement should be removed unless there is a reference for THC antitumor activity in vivo in endometrial cancer.
Response: We have to disagree with the Reviewer on this point, because the quote is directly taken from the conclusion of the study by Fonseca and colleagues where they found that THC had no effect on the survival of both human endometrial cancer cell lines. We then go on in lines 326 to 340 to qualify this observation with the lack of evidence that marijuana and consequently cannabis use has beneficial effect in endometrial cancer, whilst internet sources are using this to promote the use of untested ‘drugs’ and providing false hope to women with this disease (a key thrust of the review article). This point was identified as an article strength by Reviewer #2, and so we wanted to keep this important statement.
- Line 392, significantly vs a lot more.
Response: We have removed the phrase ‘a lot’ to make this point clearer.
- Line 444, the Author can remove all the references if they are all presented in the main body of the review.
Response: Not all of the references are included in the main text. If we removed all the references except the new reference #175, then it would appear that there is only one study worth considering here, which is clearly not the case. On this basis, we have retained the full list of pertinent references, and we hope the Referee can accept our point.
- 12. Line 464, would replace marijuana with Cannabis.
Response: We have replaced the word marijuana with cannabis, as requested.

Round 2
Reviewer 1 Report
Thank you for the response to each of my evaluations and for having considered them to modify the review